# HNF4α-TET2-FBP1 axis contributes to gluconeogenesis and type 2 diabetes

**Hongchen Li[1†], Xinchao Zhang[1,2†], Xiaoben Liang[1†], Shuyan Li[3], Ziyi Cui[1], Xinyu Zhao[1], Kai Wang[4], Bingbing Zha[4], Haijie Ma[5]\*, Ming Xu[6,7]\*, Lei Lv[2]\*, Yanping Xu[1]\***

[1]Tongji Hospital, Frontier Science Center for Stem Cell Research, Shanghai Key Laboratory of Signaling and Disease Research, School of Life Sciences and Technology, Tongji University, Shanghai, China; [2]MOE Key Laboratory of Metabolism and Molecular Medicine, Department of Biochemistry and Molecular Biology, School of Basic Medical Sciences, Fudan University, Shanghai, China; [3]Department of Radiation Oncology, Ruijin Hospital, Shanghai Jiaotong University School of Medicine, Shanghai, China; [4]Department of Endocrinology, Fifth People's Hospital of Shanghai, Fudan University, Shanghai, China; [5]Cellular and Molecular Biology Laboratory, Affiliated Zhoushan Hospital of Wenzhou Medical University, Zhoushan, China; [6]Department of Biochemistry, Molecular Biology and Biophysics, University of Minnesota, Minneapolis, United States; [7]Institute on the Biology of Aging and Metabolism, University of Minnesota, Minneapolis, United States

**\*For correspondence:**
haijie215@163.com (HM);
mixu@umn.edu (MX);
lvlei@fudan.edu.cn (LL);
yanpingxu@tongji.edu.cn (YX)

[†]These authors contributed equally to this work

**Competing interest:** The authors declare that no competing interests exist.

## eLife Assessment

Zhang et al. present **important** findings that reveal a new role for TET2 in controlling glucose production in the liver, showing that both fasting and a high-fat diet increase TET2 levels, while its absence reduces glucose production. TET2 works with HNF4α to activate the FBP1 gene upon glucagon stimulation, while metformin disrupts TET2-HNF4α interaction, lowering FBP1 levels and improving glucose homeostasis. The results are **convincing** and expand our understanding of gluconeogenesis regulation.

**Abstract** The control of gluconeogenesis is critical for glucose homeostasis and the pathology of type 2 diabetes (T2D). Here, we uncover a novel function of TET2 in the regulation of gluconeogenesis. In mice, both fasting and a high-fat diet (HFD) stimulate the expression of TET2, and *TET2* knockout impairs glucose production. Mechanistically, FBP1, a rate-limiting enzyme in gluconeogenesis, is positively regulated by TET2 in liver cells. TET2 is recruited by HNF4α, contributing to the demethylation of the *FBP1* promoter and activating its expression in response to glucagon stimulation. Moreover, metformin treatment increases the phosphorylation of HNF4α on Ser313, which prevents its interaction with TET2, thereby decreasing the expression level of FBP1 and ameliorating the pathology of T2D. Collectively, we identify an HNF4α-TET2-FBP1 axis in the control of gluconeogenesis, which contributes to the therapeutic effect of metformin on T2D and provides a potential target for the clinical treatment of T2D.

## Introduction

Loss of glucose homeostasis can lead to type 2 diabetes (T2D), characterized by persistently elevated blood glucose levels that may result in various complications, including kidney failure and neuropathy.

Since abnormal gluconeogenic activity is the primary contributor to hepatic glucose production (HGP) (*Basu et al., 2013*; *Magnusson et al., 1992*), which is the main source of increased blood glucose concentration in T2D, targeting gluconeogenesis represents a viable strategy for maintaining blood glucose homeostasis.

Fructose 1,6-bisphosphatase (FBP1) is a rate-controlling enzyme in gluconeogenesis that catalyzes the conversion of fructose 1,6-bisphosphate to fructose 6-phosphate. Notably, a point mutation in FBP1 has been reported to significantly reduce the efficacy of metformin, a first-line drug for the treatment of T2D, suggesting that FBP1 plays a major role in the therapeutic effect of metformin (*Hunter et al., 2018*). However, the specific response of FBP1 to metformin treatment remains unclear.

Ten-eleven translocation 2 (TET2) belongs to the TET family, which includes TET1, TET2, and TET3. These enzymes function as DNA dioxygenases, catalyzing the successive oxidation of 5-methylcytosine (5mC) to 5-hydroxymethylcytosine (5hmC) (*Tahiliani et al., 2009*), 5-formylcytosine, and 5-carboxycytosine (5caC) (*He et al., 2011*; *Ito et al., 2011*). Subsequently, thymine-DNA glycosylase mediates the removal of 5caC, resulting in the formation of an unmodified cytosine, which participates in DNA demethylation and gene expression regulation (*Ito et al., 2011*). Recently, it was reported that the expression of FBP1 can be regulated by promoter methylation (*Hirata et al., 2016*; *Dong et al., 2013*), leading us to hypothesize that TET2 may play a role in regulating FBP1 expression and gluconeogenesis.

Mutations in *TET2* frequently occur in various myeloid cancers. Somatic alterations in *TET2* are observed in 50% of patients with chronic myelomonocytic leukemia and are associated with poor outcomes (*Kosmider et al., 2009*). The frequency of *TET2* mutations in patients with myelodysplastic syndromes is 19% (*Delhommeau et al., 2009*). Moreover, reversing TET2 deficiency suppresses the abnormal differentiation and self-renewal of hematopoietic stem and progenitor cells and blocks leukemia progression (*Cimmino et al., 2017*). Additionally, TET2 has also been reported to repress mTORC1 and HIF signaling, thereby suppressing tumor growth in hepatocellular carcinoma and clear cell renal cell carcinoma, respectively (*He et al., 2023*; *Zhang et al., 2022*). These studies focused on the function of TET2 in cancer development; however, it is unclear whether TET2 is involved in T2D progression. In this study, we demonstrated that TET2 is recruited by HNF4α to the *FBP1* promoter, activating *FBP1* expression through demethylation, which contributes to gluconeogenesis and T2D pathology. Furthermore, we identified the HNF4α-TET2-FBP1 axis as a target of metformin treatment, suggesting that targeting this axis may represent a potential strategy for T2D management.

## Results

### TET2 contributes to gluconeogenesis and T2D

To investigate the role of TET2 in gluconeogenesis, we developed three mouse models: one subjected to overnight fasting for 16 hr prior to testing and two subjected to high-fat feeding (HFD) for 11 days and 12 weeks, respectively. The results indicated that both fasting and HFD increased the mRNA and protein levels of Tet2 in the livers of mice compared to the normal chow group (*Figure 1A–E*), suggesting that TET2 may play an important role in gluconeogenesis. To test this hypothesis, we examined the effect of TET2 on glucose output and found that TET2 overexpression promoted glucose output in HepG2 cells and primary mouse hepatocytes (*Figure 2A and B*). Consistent with this, *TET2* knockout impaired gluconeogenesis in HepG2 cells and mouse hepatocytes, even under glucagon treatment (*Figure 2C and D*). Collectively, these data demonstrated that TET2 contributes to gluconeogenesis, prompting us to investigate whether TET2 is involved in T2D progression. Pyruvate tolerance test (PTT), glucose tolerance test (GTT), and insulin tolerance test (ITT) were conducted, revealing *Tet2* KO significantly increased glucose tolerance and insulin sensitivity compared to the control mice (*Figure 2E–H*). Additionally, to assess the plasma insulin levels in response to GTT in *Tet2*-KO mice, we collected the orbital venous blood and examined the insulin levels at different time points. The results showed that *Tet2-KO* mice exhibited lower insulin secretion after glucose administration, further reflecting higher insulin sensitivity (*Figure 2H*). Meanwhile, we compared the body weight differences between wild-type (WT) and *Tet2*-KO mice and found no significant differences at 8 and 10 weeks of age on normal chow (*Figure 2I*). In summary, the loss of *Tet2* function decreased gluconeogenesis in the liver and may contribute to the treatment of T2D.

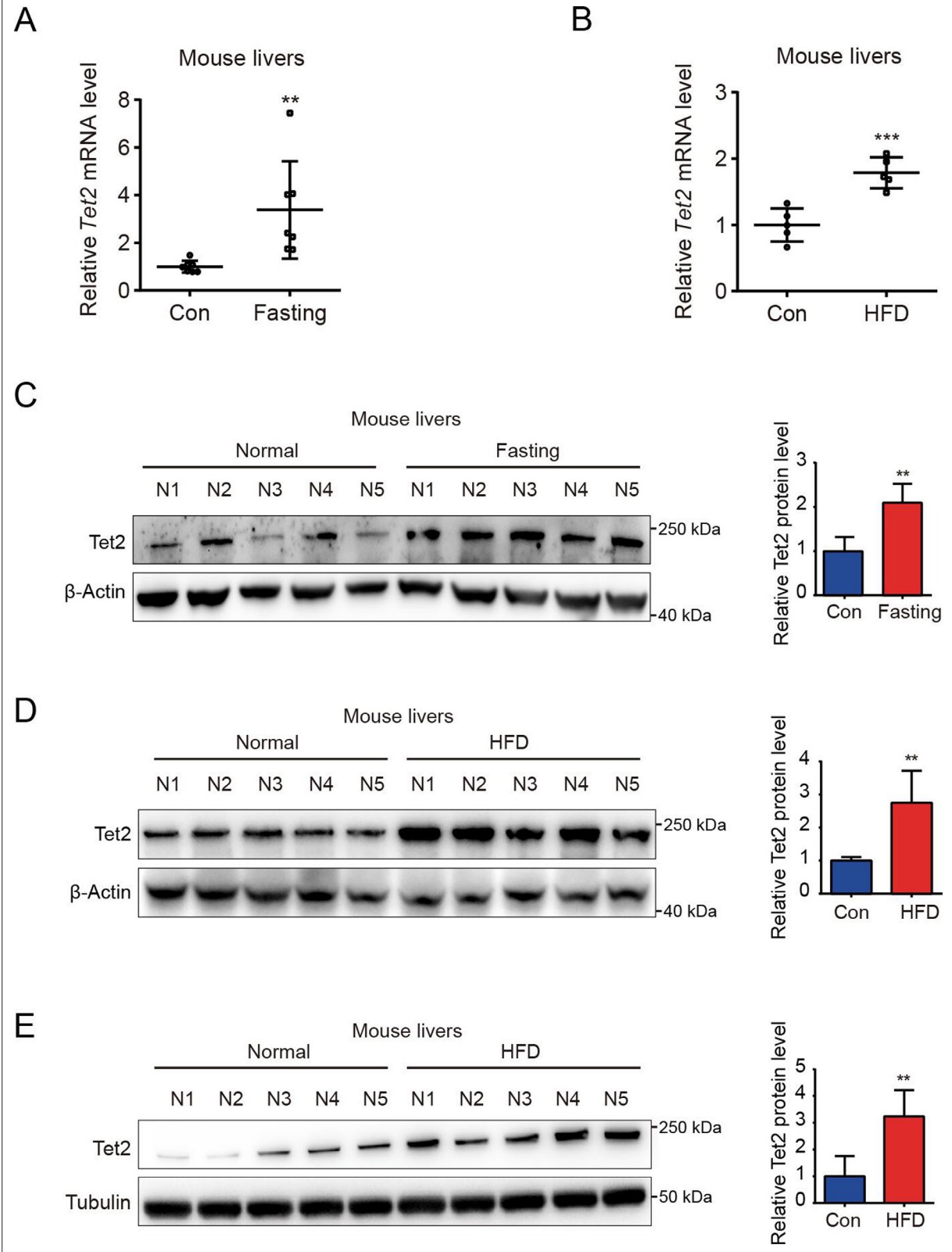

**Figure 1.** TET2 expression increases in fasting and high-fat diet (HFD) mouse livers. (**A**) qRT-PCR analysis of *Tet2* mRNA levels in mouse livers after 16 hr fasting treatment. The data are normalized to the *Actb* expression (n=7). (**B**) qRT-PCR analysis of *Tet2* mRNA levels in mouse livers after HFD treatment for 11 days. The data are normalized to the *Actb* expression (n=7). (**C**) Western blot analysis and quantification of Tet2 protein levels in mouse livers following 16 hr fasting treatment (n=5). (**D**) Western blot analysis and quantification of Tet2 protein levels in mouse livers following the HFD treatment

*Figure 1 continued on next page*

*Figure 1 continued*

for 11 days (n=5). (**E**) Western blot analysis and quantification of Tet2 protein levels in mouse livers following the 12-week HFD treatment (n=5).Statistical significance was determined using a two-tailed Student's *t*-test (**p < 0.01, ***p < 0.001). Data are represented as the mean ± SD.

The online version of this article includes the following source data for figure 1:

**Source data 1.** PDF file containing original western blots for *Figure 1C, D, E*, indicating the relevant bands and treatments.

**Source data 2.** Original files for western blot analysis displayed in *Figure 1C, D, E*.

## TET2 upregulates FBP1 expression in liver cells

Next, we explored the potential mechanism by which TET2 increases gluconeogenesis. FBP1, a rate-limiting enzyme in gluconeogenesis, plays a crucial role in T2D and was recently identified as a target of metformin (*Hunter et al., 2018*). This led us to hypothesize that FBP1 might participate in TET2-mediated regulation of gluconeogenesis. The results showed that glucagon significantly increased both TET2 and FBP1 expression levels in HepG2 and primary mouse liver cells (*Figure 3A–C*), suggesting that TET2 may regulate gluconeogenesis via FBP1. To assess the long-term effects of a single-dose glucagon, we examined *TET2* and *FBP1* mRNA levels at different times in HepG2 cells. Interestingly, the results showed that the expression peak of *TET2* and *FBP1* mRNA levels occurred 30 min after glucagon treatment, with the prolonged effects of glucagon on *TET2* mRNA lasting for more than 48 hr (*Figure 3D*), while the expression of *PEPCK* and *G6PC1* mRNA levels increased 30 min after glucagon treatment, reached the peak after 3 hr and fell to basal levels after 12 hr (*Figure 3E*), indicating the TET2-FBP1 axis may play an important role in gluconeogenesis in response to glucagon, and this regulation lasts longer than glucagon-induced *PEPCK* and *G6PC*1 expression. Furthermore, fasting and HFD also upregulated *Fbp1* mRNA levels in mouse livers (*Figure 3F and G*). Notably, Pearson correlation analysis revealed a positive correlation between *Tet2* and *Fbp1* expression in both control and fasting groups (*Figure 3H*). To confirm this, Gene Expression Profiling Interactive Analysis (GEPIA) (*Tang et al., 2017*) was used to analyze the correlation between TET2 and FBP1 expression in human liver tissue. Consistent with the mouse data, FBP1 expression levels positively correlated with TET2 levels in human livers (*Figure 3I*). These findings prompted us to examine whether TET2 regulates FBP1 expression. The results showed that TET2 overexpression promoted FBP1 expression in primary mouse hepatocytes and HepG2 cells (*Figure 3J*), while *TET2* KO significantly decreased FBP1 levels in HepG2 and LO-2 cells (*Figure 3K and L*). Moreover, *TET2* KO abolished the glucagon-induced upregulation of FBP1 (*Figure 3M*), suggesting that TET2 is required for glucagon-induced FBP1 upregulation. Taken together, these data indicate that TET2 regulates FBP1 expression in liver cells.

To explore the mechanism by which TET2 regulates FBP1, we performed ChIP-qPCR to investigate whether TET2 binds to the FBP1 promoter and catalyzes the conversion of 5mC to 5hmC in HepG2 cells under glucagon treatment. We found that glucagon treatment promoted TET2 binding to the FBP1 promoter in HepG2 cells, increasing 5hmC levels and reducing 5mC levels (*Figure 3N–P*). Importantly, *TET2* knockout blocked this process and led to the accumulation of 5mC in the FBP1 promoter (*Figure 3N–P*). These data demonstrate that TET2 mediates the transcriptional activation of FBP1 in response to glucagon stimulation.

## HNF4α is required for TET2-mediated transcriptional activation of FBP1

Given that TET2 binds to DNA without sequence specificity, we sought to understand how TET2 specifically activates FBP1 expression in response to glucagon treatment. Recent genome-wide methylation and transcriptome analyses identified hepatocyte nuclear factor 4 alpha (HNF4α) as a master gluconeogenic transcription factor that plays a critical role in the pathogenesis of diabetic hyperglycemia (*Zhang et al., 2018*). Importantly, ChIP-seq data suggest that HNF4α binds to the FBP1 promoter and participates in the regulation of gluconeogenesis in adult mouse hepatocytes (*Rhee et al., 2003*; *Alder et al., 2014*). To determine whether HNF4α is involved in TET2-mediated FBP1 expression, we performed immunofluorescence to examine the co-localization of TET2 and HNF4α with or without glucagon treatment in HepG2 cells. The results indicated that the co-localization of TET2 and HNF4α significantly increased upon glucagon treatment (*Figure 4A*). Moreover, fasting and HFD treatments also promoted the co-localization of Tet2 and Hnf4α in mouse hepatocyte cells (*Figure 4B and C*). Consistently, we observed that glucagon increased the interaction between TET2

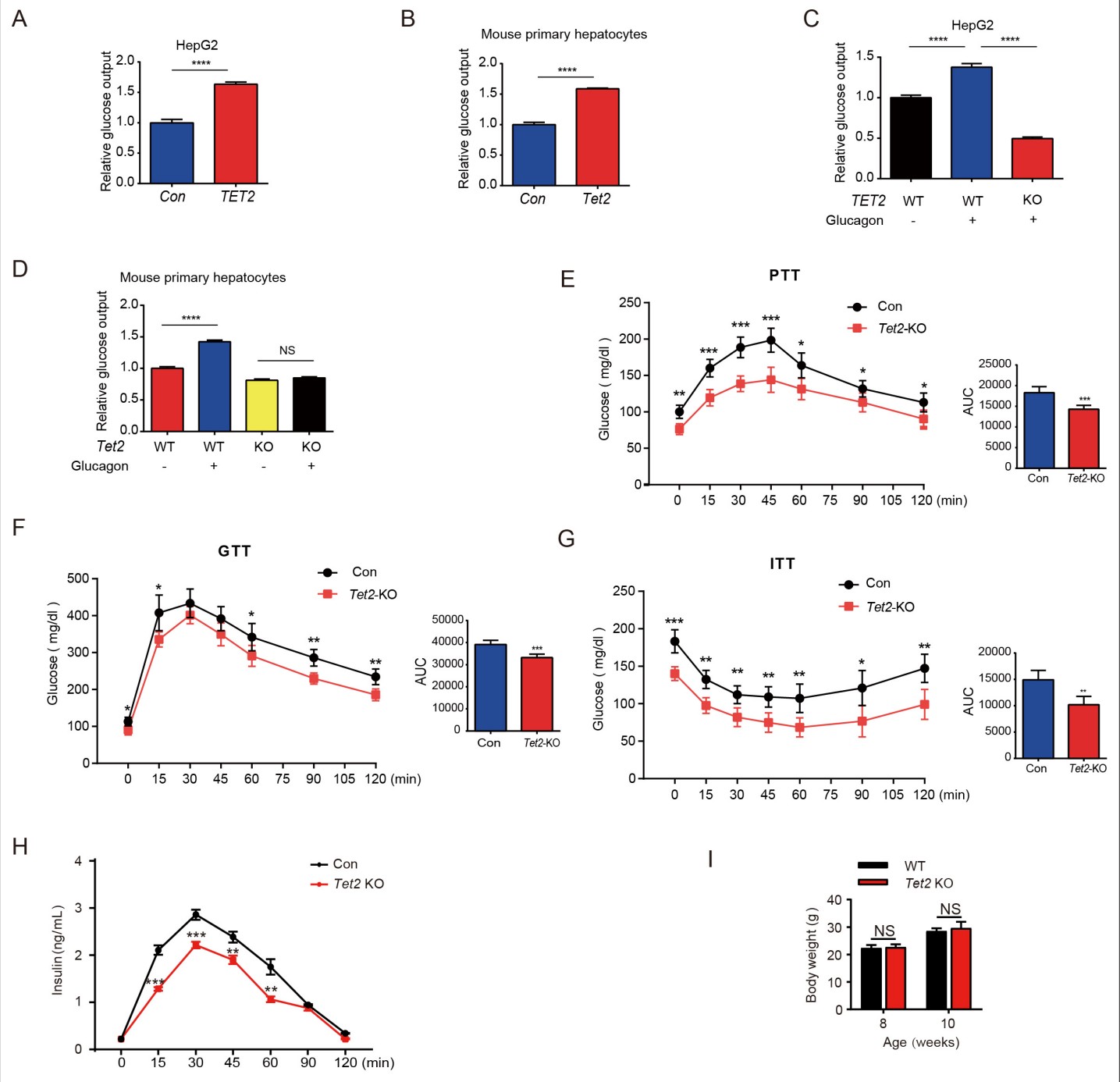

**Figure 2.** TET2 boosts gluconeogenesis. (**A**) Glucose production assays were performed in HepG2 cells after TET2 overexpression. Data are represented as the mean ± SD (n=3). (**B**) Glucose production assays were performed in mouse primary hepatocytes after *Tet2* overexpression. Data are represented as the mean ± SD (n=3). (**C**) Glucose production assays were performed in HepG2 cells pre-treated with 20 nM glucagon in wild-type (WT) and *TET2* KO HepG2 cells. Data are represented as the mean ± SD (n=3). (**D**) Glucose production assays were performed in mouse primary hepatocytes pre-treated with 20 nM glucagon in WT and *Tet2* KO cells. Data are represented as the mean ± SD (n=3). (**E**) Pyruvate tolerance test (PTT) was performed following a 16 hr fasting treatment and intraperitoneal (i.p.) injection of 1 g/kg sodium pyruvate (n = 5). (**F**) Glucose tolerance test (GTT) was performed after a 12 hr fasting treatment and i.p. injection of 2 g/kg glucose (n = 5). (**G**) Insulin tolerance test (ITT) was performed after a 4 hr fasting treatment and i.p. injection of 0.75 U/kg insulin (n = 5). (**H**) Glucose-stimulated insulin secretion was examined. After fasting and i.p. injection of 2 g/kg glucose, plasma insulin levels were measured at the indicated time points (n = 5). (**I**) Body weight of 8- or 10-week-old male WT mice and *Tet2* KO mice on a normal chow diet (n = 5). Statistical significance was determined using a two-tailed Student's *t*-test (*p < 0.05, **p < 0.01, ***p < 0.001 , and ****p < 0.0001).

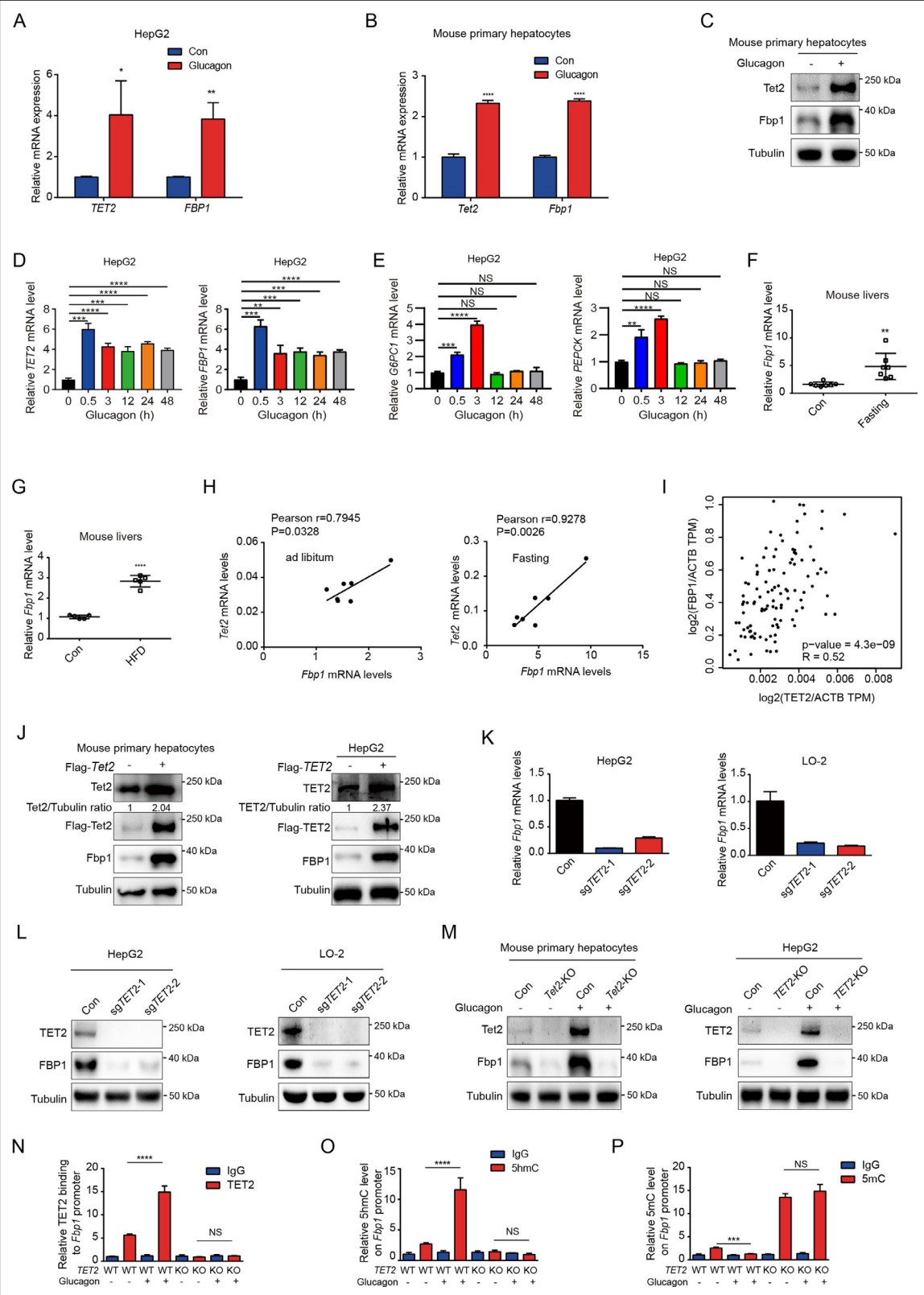

**Figure 3.** TET2 upregulates FBP1 expression in liver cells. (**A**) qPCR analysis of *TET2* and *FBP1* mRNA expression levels after 20 nM glucagon treatment for 48 hr in HepG2 cells. Data are represented as the mean ± SD (n=3). Statistical significance was determined using a two-tailed Student's *t*-test (*p < 0.05, **p < 0.01). (**B**) qPCR analysis of *Tet2* and *Fbp1* mRNA expression levels after 20 nM glucagon treatment for 48 hr in primary mouse hepatocytes. Data are represented as the mean ± SD (n=3). Statistical significance was determined using a two-tailed Student's *t*-test (****p < 0.0001). (**C**) Western

*Figure 3 continued on next page*

*Figure 3 continued*

blot analysis of Tet2 and Fbp1 protein levels after 20 nM glucagon treatment in mouse primary hepatocyte cells. (**D**) qPCR analysis of *TET2* and *FBP1* mRNA expression levels after 20 nM glucagon treatment at the indicated time points (n = 3). Statistical significance was determined using a two-tailed Student's *t*-test (**p < 0.01, ***p < 0.001 and ****p < 0.0001). (**E**) qPCR analysis of *G6PC1* and *PEPCK* mRNA expression levels after 20 nM glucagon treatment at the indicated time points in HepG2 cells (n = 3). Statistical significance was determined using a two-tailed Student's *t*-test (**p < 0.01,, ***p < 0.001 and ****p < 0.0001). (**F**) qPCR analysis of *Fbp1* mRNA levels in mouse livers following fasting treatment (n=7). Statistical significance was determined using a two-tailed Student's *t*-test (**p < 0.01). (**G**) qPCR analysis of *Fbp1* mRNA levels in mouse livers following high-fat diet (HFD) treatment (n=5). Statistical significance was determined using a two-tailed Student's *t*-test (****p < 0.0001). (**H**) Correlation analysis between *TET2* and *FBP1* levels using data from *Figure 1A* in mouse livers with or without fasting treatment. (**I**) Correlation analysis between *TET2* and *FBP1* levels in human livers. Data were collected from Gene Expression Profiling Interactive Analysis (GEPIA) (*Tang et al., 2017*). (**J**) Western blot analysis of TET2 and FBP1 expression after overexpression of Flag-TET2 in mouse primary hepatocytes and HepG2 cells. (**K**) qPCR analysis of *FBP1* expression levels in control and *TET2* knockout HepG2 and LO-2 cells. (**L**) Western blot analysis of TET2 and FBP1 protein levels in control and *TET2* knockout HepG2 and LO-2 cells. (**M**) Western blot analysis of Tet2 and Fbp1 protein levels in control and *Tet2* knockout mouse primary hepatocytes and HepG2 cells treated with or without 20 nM glucagon. (**N**) ChIP-qPCR analysis of TET2 binding to *FBP1* promoter in response to glucagon stimulation in control and *TET2* knockout HepG2 cells. (**O**) ChIP-qPCR analysis of 5-hydroxymethylcytosine (5hmC) levels in *FBP1* promoter in response to glucagon stimulation in control and *TET2* knockout HepG2 cells. (**P**) ChIP-qPCR analysis of 5-methylcytosine (5mC) levels in *FBP1* promoter in response to glucagon stimulation in control and *TET2* knockout HepG2 cells (n=3). N-P are presented as the mean ± SD using a Tukey's post hoc test (***p < 0.001 and ****p < 0.0001).

The online version of this article includes the following source data for figure 3:

**Source data 1.** PDF file containing original western blots for *Figure 3C, J, L and M*, indicating the relevant bands and treatments.

**Source data 2.** Original files for western blot analysis displayed in *Figure 3C, J, L and M*.

and HNF4α in HepG2 cells (*Figure 4D*). To determine the function of HNF4α in TET2-mediated regulation of FBP1 expression, we knocked down *HNF4A* in HepG2 cells using siRNA (*Figure 4E*) and found that *HNF4A* knockdown significantly inhibited glucagon-induced TET2 binding to the *FBP1* promoter (*Figure 4F*), decreased 5hmC levels (*Figure 4G*), and increased 5mC levels in the *FBP1* promoter (*Figure 4H*), thereby suppressing TET2-mediated FBP1 expression (*Figure 4I*) and impairing glucose output under glucagon treatment (*Figure 4J*), demonstrating that HNF4α recruits TET2 to the *FBP1* promoter and activates FBP1 expression through demethylation to facilitate gluconeogenesis. Notably, the expression of both *Hnf4a* and *Fbp1* also increased in mouse livers under fasting or HFD treatment compared to the control group (*Figure 4K–M*). In conclusion, these results support the notion that TET2-mediated FBP1 expression and gluconeogenesis is dependent on HNF4α.

## HNF4α phosphorylation affects its binding to TET2 and FBP1 expression

Our results demonstrated that HNF4α recruits TET2 to the FBP1 promoter and activates FBP1 expression through demethylation, playing a crucial role in the regulation of hepatic glucose output. However, it remains unclear whether the HNF4α-TET2-FBP1 axis responds to metformin treatment. Metformin, a first-line antidiabetic drug widely used to treat hyperglycemia in T2D, is also a well-established adenosine 5'-monophosphate-activated protein kinase (AMPK) activator (*Aroda et al., 2017*). Interestingly, one study revealed that AMPK phosphorylates HNF4α at Ser 313, reducing its transcriptional activity (*Hong et al., 2003*). Combining these studies with our findings, we wonder whether metformin can affect the HNF4α-TET2-FBP1 axis. To explore the role of metformin-induced AMPK phosphorylation of HNF4α in FBP1 expression, we treated cells with metformin and assessed the interaction between HNF4α and TET2. The results showed that metformin administration impaired HNF4α's ability to bind to TET2 (*Figure 5A*), leading to a significant reduction in TET2 binding to the FBP1 promoter (*Figure 5B*). Consistently, metformin treatment significantly decreased the expression level of FBP1 (*Figure 5C*). Notably, metformin also induced high levels of HNF4α phosphorylation at Ser 313 (*Figure 5C*). To determine the effect of HNF4α phosphorylation on the HNF4α-TET2-FBP1 axis, we transfected HepG2 cells with WT HNF4α and Ser 313 mutants. The results showed that the phosphomimetic mutation (S313D) of HNF4α impaired its ability to bind to TET2 (*Figure 5D*), prevented TET2 from binding to the FBP1 promoter (*Figure 5E*), and reduced FBP1 expression at both mRNA and protein levels (*Figure 5F*). In contrast, the phosphoresistant mutation (S313A) showed higher activity in interacting with TET2, recruiting TET2 to the FBP1 promoter, and activating its expression (*Figure 5D–F*). Taken together, these data demonstrate that metformin-mediated HNF4α phosphorylation suppresses FBP1 expression by preventing TET2 recruitment to the FBP1 promoter by HNF4α.

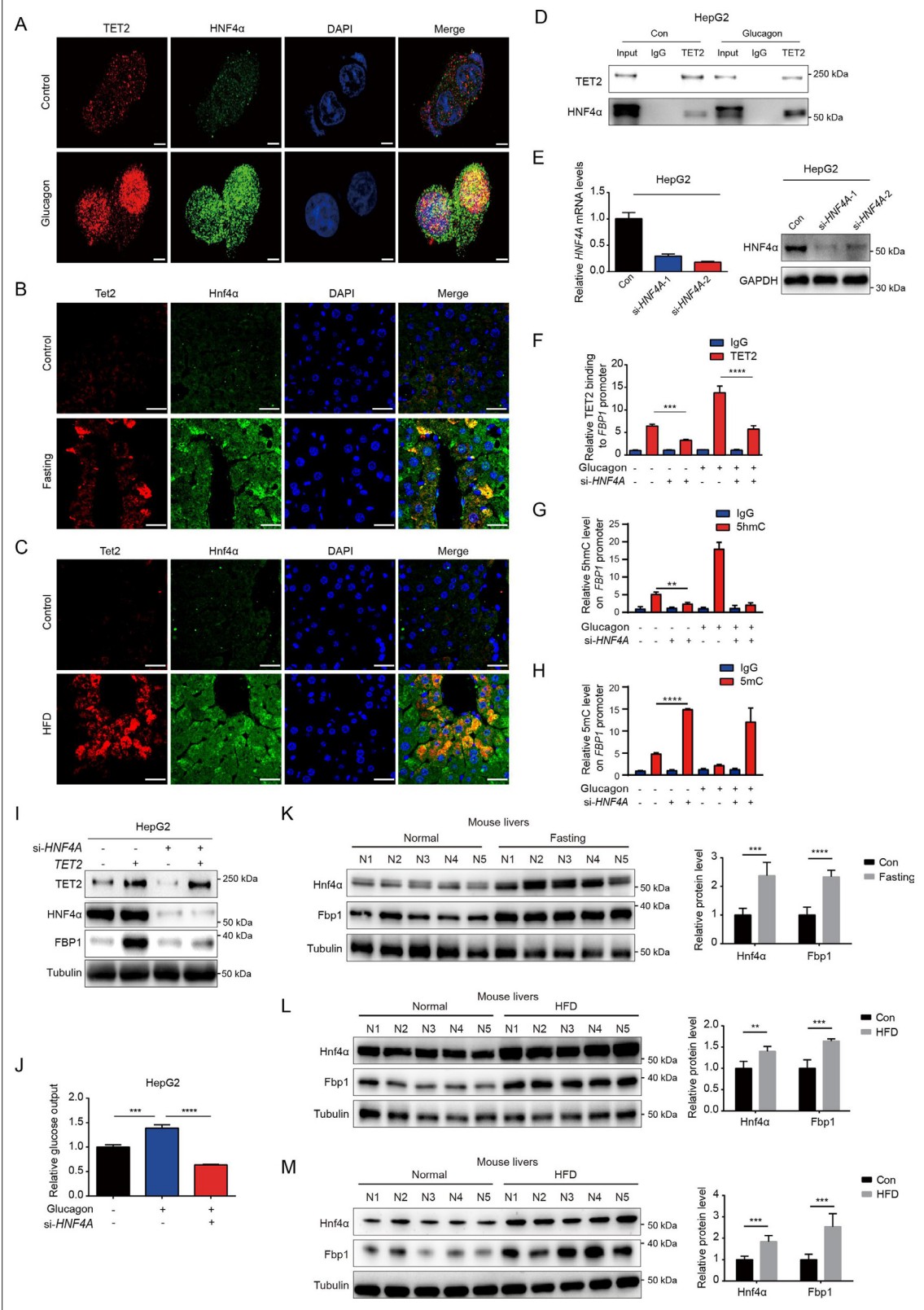

**Figure 4.** HNF4α is necessary for TET2-mediated FBP1 upregulation. (**A**) Immunofluorescence analysis of TET2 and HNF4α co-localization in HepG2 cells after 20 nM glucagon treatment for 48 hr. Scale bar: 10 μm. (**B**) Immunofluorescence analysis of Tet2 and Hnf4α co-localization in liver sections from standard chow and fasting mice. Scale bar: 30 μm. (**C**) Immunofluorescence analysis of Tet2 and Hnf4α co-localization in liver sections from standard chow and high-fat diet (HFD) mice. Scale bar: 30 μm. (**D**) Endogenous co-immunoprecipitation followed by western blot analysis of the interaction

*Figure 4 continued on next page*

*Figure 4 continued*

between HNF4α and TET2 with or without glucagon treatment in HepG2 cells. (**E**) qRT-PCR and western blot analysis of HNF4α expression levels in HepG2 cells transfected with two specific siRNAs. (**F**) ChIP-qPCR analysis of TET2 binding to *FBP1* promoter in HepG2 cells treated with siRNA targeting *HNF4A* and glucagon as indicated (n=3). Statistical significance was determined using a two-tailed Student's *t*-test (***p < 0.001, ****p < 0.0001). (**G**) ChIP-qPCR analysis of 5-hydroxymethylcytosine (5hmC) levels in the *FBP1* promoter in HepG2 cells treated with siRNA targeting *HNF4A* and glucagon as indicated (n=3). Statistical significance was determined using a two-tailed Student's *t*-test (**p < 0.01). (**H**) ChIP-qPCR analysis of 5-methylcytosine (5mC) levels in the *FBP1* promoter in HepG2 cells treated with siRNA targeting *HNF4A* and glucagon as indicated (n=3). Statistical significance was determined using a two-tailed Student's *t*-test (****p < 0.0001). (**I**) Western blot analysis of FBP1 protein levels in HepG2 cells treated with TET2 overexpression and siRNA targeting *HNF4A* as indicated. (**J**) Glucose production assays were performed in HepG2 cells treated with glucagon and transfected with *HNF4A* siRNA as indicated (n=3). Statistical significance was determined using a two-tailed Student's *t*-test (***p < 0.001, ****p < 0.0001). (**K–M**) Western blot analysis and quantification of Hnf4a and Fbp1 protein levels in mouse livers from the mice treated with 16 hr overnight fasting (**K**) or 11-day HFD (**L**), or 12-week HFD treatment (**M**). n=5. Statistical significance was determined using a two-tailed Student's *t*-test (**p < 0.01, ***p < 0.001 and ****p < 0.0001).

The online version of this article includes the following source data for figure 4:

**Source data 1.** PDF file containing original western blots for *Figure 4D, E, I, K, L and M*, indicating the relevant bands and treatments.

**Source data 2.** Original files for western blot analysis displayed in *Figure 4D, E, I, K, L and M*.

## Targeting TET2 improves the efficacy of metformin in glucose metabolism in vivo

Increased rates of gluconeogenesis, derived from continuously excessive HGP, result in abnormal glucose homeostasis in T2D. Fasting or HFD-induced T2D mice showed elevated levels of *Tet2*

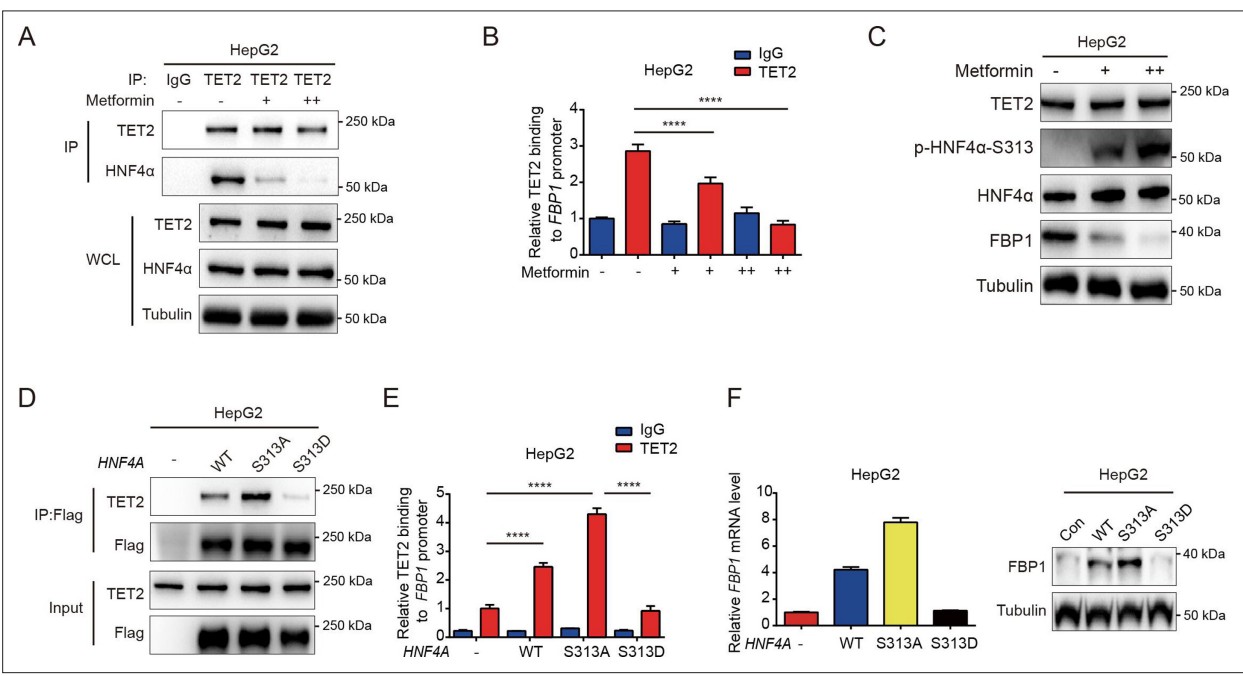

**Figure 5.** Metformin impairs the ability of HNF4α binding to TET2 and FBP1 expression. (**A**) Endogenous co-immunoprecipitation followed by western blot analysis of the interaction between HNF4α and TET2 with or without metformin (10 mM) treatment in HepG2 cells. (**B**) ChIP-qPCR analysis of TET2 binding to *FBP1* promoter in HepG2 cells treated with or without metformin (10 mM) (n=3). Statistical significance was determined using a two-tailed Student's *t*-test (****p < 0.0001). (**C**) Western blot analysis of TET2, HNF4α, HNF4α phosphorylation at Ser 313 and FBP1 levels in HepG2 cells treated with metformin as indicated (+: 5 mM, ++: 10 mM). (**D**) Endogenous co-immunoprecipitation followed by western blot analysis of the interaction between TET2 and HNF4α wild-type and S313 mutants as indicated. (**E**) ChIP-qPCR analysis of TET2 binding to *FBP1* promoter in HepG2 cells transfected with HNF4α wild-type and S313 mutants as indicated (n=3). Statistical significance was determined using ausing a Tukey's post hoc test (****p < 0.0001). (**F**) qRT-PCR and western blot analysis of FBP1 levels in HepG2 cells transfected with HNF4α wild-type and S313 mutants as indicated.

The online version of this article includes the following source data for figure 5:

**Source data 1.** PDF file containing original western blots for *Figure 5A, C, D and F*, indicating the relevant bands and treatments.

**Source data 2.** Original files for western blot analysis displayed in *Figure 5A, C, D and F*.

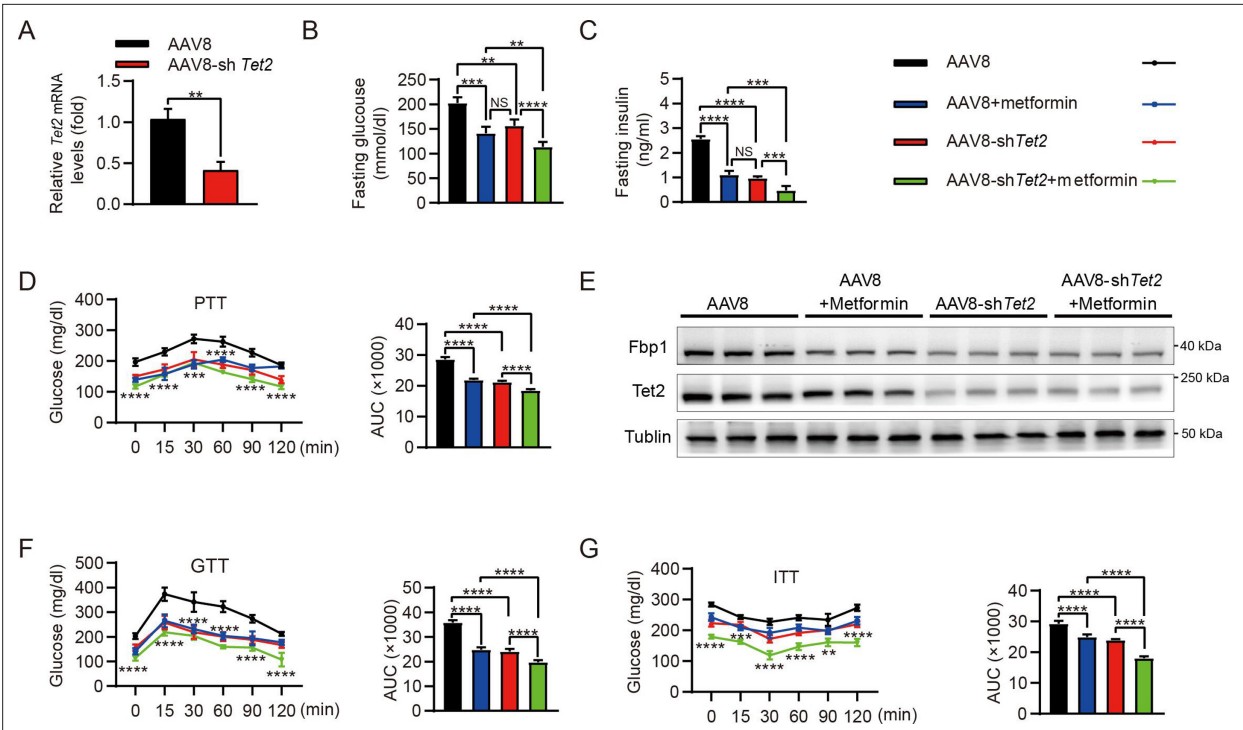

**Figure 6.** Targeting TET2 improves the efficacy of metformin in glucose metabolism in vivo. (**A**) qRT-PCR analysis of *Tet2* mRNA levels in mouse livers from the high-fat diet (HFD) mice infected with AAV8 or AAV8-*shTet2* for 10 days (n=6). Statistical significance was determined using a two-tailed Student's *t*-test (**p < 0.01) . (**B, C**) Analysis of fasting blood glucose (**B**) and plasma insulin (**C**) levels in the HFD mice infected with AAV8 or AAV8-*shTet2* for 10 days, and treated with or without metformin (300 mg/kg/day) for another 10 days as indicated. n=6. Statistical significance was determined using ausing a Tukey's post hoc test (**p < 0.01, ***p < 0.001, and ****p < 0.0001). (**D**) Pyruvate tolerance test (PTT) was performed in the HFD mice infected with AAV8 or AAV8-*shTet2* for 10 days, and treated with or without metformin (300 mg/kg/day) for another 10 days as indicated (n=6). (**E**) Western blot analysis of Tet2 and Fbp1 levels in livers from the HFD mice infected with AAV8 or AAV8-*shTet2* for 10 days, and treated with or without metformin (300 mg/kg/day) for another 10 days as indicated. (**F, G**) Glucose tolerance test (GTT) (**F**) and insulin tolerance test (ITT) (**G**) were performed in the HFD mice infected with AAV8 or AAV8-*shTet2* for 10 days and treated with or without metformin (300 mg/kg/day) for another 10 days as indicated (n=6). (**D, F** and **G**) are determined using a two-tailed Student's *t*-test for left panel and a Tukey's post hoc test for right panel in each figure (**p < 0.01, ***p < 0.001, and ****p < 0.0001).

The online version of this article includes the following source data for figure 6:

**Source data 1.** PDF file containing original western blots for **Figure 6E**, indicating the relevant bands and treatments.

**Source data 2.** Original files for western blot analysis displayed in **Figure 6E**.

and *Fbp1* in hepatocytes (**Figure 1A–E**). However, it remains unclear whether knockdown of *Tet2*, in combination with metformin, would decrease glucose production, thereby improving T2D treatment outcomes. HFD-induced diabetic mice were infused with AAV8 or AAV8-sh*Tet2*. To confirm the efficiency of liver-specific *Tet2* knockdown in vivo, we examined *Tet2* mRNA levels in mouse livers and found that TET2 expression significantly decreased upon AAV8-sh*Tet2* treatment (**Figure 6A**). Notably, fasting blood glucose and insulin levels were markedly lower in the AAV8-sh*Tet2* group than the control group in response to metformin treatment (**Figure 6B and C**). PTT performed on HFD mice indicated that Tet2 suppression sharply decreased hepatic gluconeogenesis (**Figure 6D**), which was consistent with lower protein levels of Fbp1 in mouse livers (**Figure 6E**). Moreover, GTT and ITT showed that *Tet2* knockdown in HFD mice significantly enhanced glucose tolerance and insulin sensitivity compared to the control group (**Figure 6F and G**). Of note, the beneficial effects of *Tet2* knockdown alone were comparable to those of metformin in lowering glucose levels and improving insulin sensitivity. Interestingly, the combination of metformin and *Tet2* knockdown in HFD mice exhibited better glucose-lowering effects and insulin sensitivity compared to either *Tet2* knockdown or metformin. Collectively, these data demonstrated that suppressing TET2 synergizes with metformin to lower glucose production and enhance insulin sensitivity by inhibiting FBP1 expression.

## Discussion

Our study revealed the function of TET2 in the regulation of gluconeogenesis. Notably, gluconeogenesis levels in *TET2* knockout HepG2 cells and primary mouse hepatocytes significantly decreased even under glucagon treatment, demonstrating that TET2 is essential for glucagon-induced upregulation of glucose output. Our findings link the novel function of TET2 to the gluconeogenic process. However, the precise role of TET2 in the pathophysiology of T2D remains unclear and requires further research. Additionally, the mechanisms by which TET2 is upregulated during gluconeogenesis deserve further exploration.

A recent study showed that metformin treatment can reduce HGP by targeting FBP1 (*Hunter et al., 2018*), suggesting that FBP1 may be a promising target for diabetes management. However, the regulation of FBP1 remains poorly understood, despite several studies indicating that promoter methylation mediates FBP1 expression silencing in cancer (*Dong et al., 2018*; *Chen et al., 2011*). Here, we found that TET2 positively regulates FBP1 in response to glucagon treatment, indicating that targeting TET2 may represent a promising strategy for treating T2D. Importantly, glucagon-induced FBP1 expression lasts longer than *PEPCK* and *G6PC1* expression. This divergent response was mechanistically linked to TET2-dependent transcriptional regulation of FBP1.

We further investigated how TET2 specifically regulates FBP1 expression in response to glucagon stimulation. Like most chromatin-modifying enzymes, which require DNA sequence-specific binding proteins, such as DNA transcription factors, for recruitment and regulation of specific gene expression (*Smith and Shilatifard, 2010*), TET2 also requires binding partners to modulate particular pathways in a context-dependent manner. This is supported by findings that Wilms' tumor protein (*Wang et al., 2015*) and Smad nuclear interacting protein 1 (*Chen et al., 2018*) are indispensable for TET2 to suppress leukemia cell proliferation and regulate the cellular DNA damage response, respectively. Our data demonstrated that TET2 upregulates FBP1, thereby enhancing gluconeogenesis in an HNF4α-dependent manner. Furthermore, metformin-induced phosphorylation of HNF4α at Ser 313 impairs its binding ability to TET2 and reduces TET2 recruitment to the FBP1 promoter, leading to decreased FBP1 expression. Importantly, TET2-FBP1 inhibition has a synergistic effect with metformin in HFD mice.

Intriguingly, a clinical investigation study revealed that *TET2* mutations occur more frequently in the diabetes mellitus (DM) group than the non-DM, suggesting a potential connection between TET2 and insulin resistance (*Xu et al., 2024*). Additionally, inactivating mutations of the epigenetic regulator TET2 led to metabolic dysfunction, including clonal hematopoiesis, and aggravated age- and obesity-related insulin resistance in mice (*Fuster et al., 2020*). Regarding HNF4α variants, an analysis utilizing exome sequencing data demonstrated that human genetic variations in HNF4α disrupted its protein structure and function, impaired insulin secretion, reduced sensitivity to insulin, and increased the risk of T2D in individuals (*Ellard and Colclough, 2006*; *Yamagata et al., 1996*). Furthermore, it has been reported that a point mutation in FBP1 can reduce the efficacy of metformin treatment (*Hunter et al., 2018*), providing genetic evidence for the role of the TET2-FBP1 axis in the therapeutic effect of metformin. In summary, our findings uncovered a previously unknown function of TET2 in gluconeogenesis. TET2, together with HNF4α, facilitates FBP1 expression by maintaining the hypomethylation of the FBP1 promoter, which leads to increased gluconeogenesis. Thus, targeting the HNF4α-TET2-FBP1 axis may represent a promising strategy to lower blood glucose in T2D.

## Materials and methods
### Cell culture and transfection

HepG2 (#iCell-h092), LO-2 (#iCell-h054), and HEK293T (#iCell-h243) were purchased from Cellverse Co., Ltd (Shanghai, China), and have been confirmed to be free of mycoplasma contamination. Moreover, cell lines and primary mouse hepatocytes were cultured in DMEM supplemented with 10% fetal bovine serum and incubated at 37°C in a 5% $CO_2$ atmosphere. For plasmids or siRNA transfection, FuGENE HD (#E2311, Roche, WI, USA) and Lipofectamine 2000 (#11668030, Invitrogen, CA, USA) were utilized, respectively. For lentivirus production, polyethyleneimine from EZ Trans (#AC04L091, Heyuan Liji, Shanghai, China) was used. All transfection procedures were conducted according to the manufacturer's instructions. The sequences of *HNF4A* siRNA are listed as follows:

5'-AAUGUAGUCAUUGCCUAGGTT-3' and
5'-UCUUGUCUUUGUCCACCACTT-3'.

## Isolation of primary mouse hepatocytes

Primary mouse hepatocytes were isolated by perfusing the liver with pre-warmed Hank's Balanced Salt Solution at a flow rate of 5–7 mL/min for 5 min after anesthetizing the mouse. Once the liver appeared pale, the perfusion solution was switched to a digestion buffer, maintaining the same flow rate and duration. The liver was then transferred to a 10 cm dish containing 10 mL of digestion medium. Hepatocytes were released by gently cutting and agitating the liver, after which 20 mL of cold phosphate-buffered saline (PBS) was added to halt the trypsin digestion. The cell suspension was filtered through a 70 µm nylon mesh (BD Falcon), centrifuged at $50 \times g$ for 2 min, and the cell pellet was washed twice with cold PBS at $50 \times g$ for 5 min each. The hepatocytes were then ready for subsequent experiments.

## Western blot and immunoprecipitation

Cell lysates were prepared using NP40 lysis buffer, and proteins were separated by SDS-PAGE. The following primary antibodies were used: Tubulin (#66031-1-Ig, Proteintech, Wuhan, China; 1:1000), TET2 (#18950S, CST, MA, USA; 1:1000), HNF4α (#32591, SAB, MD, USA; 1:1000), HNF4α-pS313 (ab78356, Abcam, Cambridge, UK, 1:1000), FBP1 (#HPA005857, Sigma, MO, USA; 1:1000), and HA (#3724, CST, MA, USA; 1:1000). Secondary antibodies included anti-rabbit (#L3012, SAB, MD, USA; 1:1000) and anti-mouse (#L3032, SAB, MD, USA; 1:1000). For immunoprecipitation, total protein was incubated with protein A beads (#37484, CST, MA, USA) and TET2 antibody (#MABE 462, Millipore, MO, USA; 1:1000) for 3 hr at 4°C, followed by western blot analysis.

## Reverse transcription qPCR

Total RNA was extracted using an RNA purification kit (#B0004D, EZBioscience, MN, USA). Real-time PCR was conducted using SYBR Green (#11200ES03, YEASEN, Shanghai, China) following cDNA synthesis. *ATCB* and *Atcb* were used as an internal control. The sequences of the qPCR primers are as follows:

> *hTET2*-forward: GATAGAACCAACCATGTTGAGGG;
> *hTET2*-reverse: TGGAGCTTTGTAGCCAGAGGT;
> *hFBP1*-forward: CGCGCACCTCTATGGCATT;
> *hFBP1*-reverse: TTCTTCTGACACGAGAACACAC;
> *hPEPCK*-forward: AGACCAACCTGGCCATGATG;
> *hPEPCK*-reverse: GCGACACCGAAAAAGCCATT;
> *hG6PC1*-forward: GACAGCGTCCATACTGGTGG;
> *hG6PC1*-reverse: GTATACACCTGCTGTGCCCAT;
> *hACTB*-forward: CATGTACGTTGCTATCCAGGC;
> *hACTB*-reverse: CTCCTTAATGTCACGCACGAT;
> *mTet2*-forward: AGAGAAGACAATCGAGAAGTCGG;
> *mTet2*-reverse: CCTTCCGTACTCCCAAACTCAT;
> *mFbp1*-forward: CACCGCGATCAAAGCCATCT;
> *mFbp1*-reverse: CCAGTCACATTGGTTGAGCCA;
> *mActb*-forward: GTGACGTTGACATCCGTAAAGA;
> *mActb*-reverse: GCCGGACTCATCGTACTCC.

## Chromatin immunoprecipitation

Chromatin immunoprecipitation (ChIP) assay was performed based on the manufacturer's instructions (#P2078, Beyotime Biotechnology, Shanghai, China). Briefly, HepG2 cells were washed with PBS and cross-linked using formaldehyde of final concentration of 1% (Sigma, USA) for 10 min, at room temperature. Genomic DNA of HepG2 cell lysate was digested with micrococcal nuclease (#D7201S, Beyotime Biotechnology, Shanghai, China) for 20 min at 37°C, and then sonicated to produce nucleotide fractions of 200–1000 base pairs. The mixed solution was centrifuged at 13,000 rpm at 4°C to separate the sediments. The sediments were resuspended with ChIP dilution buffer (#P2078, ChIP

Assay Kit, Beyotime Biotechnology, Shanghai, China) and incubated with anti-TET2 antibody (#18950S, CST, MA, USA; 1:1000), anti-5mC antibody (#HA601350, HUABIO, Hangzhou, China), anti-5hmC antibody (#HA601351, HUABIO, Hangzhou, China), and anti-IgG primary antibody (#10284-1-AP, Proteintech, Wuhan, China) at 4°C overnight. The mixture was incubated with Protein A/G Sepharose beads (#HY-K0230, MCE, Shanghai, China) for 2 hr at 4°C, followed by centrifugation at 1000 rpm at 4°C. The antibody-conjugated Protein A/G Sepharose beads were washed with PBS and then eluted with fresh elution buffer to dissociate the conjugated DNA. DNA was isolated using a DNA purification kit (#B518141, Sangon Biotech, Shanghai, China), followed by ChIP-PCR. The primers used for amplification of immunoprecipitated DNA are as follows:

FBP1-Forward 5′-GATCCCCGACCTTGTCTGAA-3′;
FBP1-Reverse 5′-TCGCGGAAACCTTTAGACGC-3′.

## Gene knockout cells and mutagenesis

The CRISPR/Cas9 system was employed to generate TET2 knockout cells. Lentiviruses were produced by co-transfecting HEK293T cells with plasmids containing sgRNA (8 µg), psPAX2 (6 µg), and pMD2.G (2 µg) in a 10 cm dish. The collected supernatant was used to infect the target cells. Following selection with puromycin, western blot analysis was performed to assess the efficiency of the knockout. The targeting sequence for TET2 sgRNA was 5′-GATTCCGCTTGGTGAAAACG-3′. For mutagenesis, human HNF4α cDNA was cloned into a pLVX-2×Flag lentiviral expression vector. Both site-directed mutants of HNF4α, including HNF4α-S313A and HNF4α-S313D, were generated by PCR using KOD FX (#KFX-201, TOYOBO, Japan) following the manufacturer's protocol. Briefly, HNF4α plasmids were amplified, and the products were digested with DpnI enzyme (#R0176V, New England Biolabs, MA, USA) before being transformed into NcmDH5-α (#MD101-1, NCM Biotech, Suzhou, China) for amplification.

## Immunofluorescence

For immunofluorescence, the HepG2 cells or mouse liver sections were incubated with related primary antibodies overnight at 4°C. Mouse anti-TET2 (#ANM4475, IPODIX, Wuhan, China; 1:100) and rabbit anti-HNF4α (#ET1611-43, HUABIO, Hangzhou, China; 1:100) were used. The cells or sections were rinsed with PBS three times, then incubated with Alexa Fluor 488-conjugated or Alexa Fluor 555-conjugated secondary antibodies (#A0460, # A0423, Beyotime, Shanghai, China; 1:200) for 2 hr at room temperature. The HepG2 cells or sections were rinsed with PBS three times and stained with ProLong Diamond Antifade Mountant with DAPI solution (#P36971, Invitrogen, CA, USA), and coated with coverslips (#174950, Invitrogen, CA, USA). All images were captured by using a confocal microscope and processed with Fluoview software (Olympus-FV3000, Tokyo, Japan), ImageJ, and Photoshop CS5.

## Glucose production

Cells were washed three times with PBS before the medium was replaced with glucose-free DMEM (#A14430-01, Gibco, CA, USA) supplemented with 20 mM sodium lactate and 2 mM sodium pyruvate. After an 8 hr incubation, glucose levels in the supernatants were measured using a glucose assay kit (#A22189, Thermo Fisher, MA, USA).

## PTT, GTT, and ITT

For the PTT and GTT, mice were fasted for 16 hr and 12 hr, respectively, before receiving an intraperitoneal (i.p.) injection of either pyruvate (2 g/kg) or glucose (2 g/kg). In the ITT, mice fasted for 4 hr in the morning prior to receiving an insulin injection (1 U/kg, i.p.). Blood glucose levels were measured at intervals of 0, 15, 30, 45, 60, 90, and 120 min after injection using a glucometer during tail vein bleeding.

## Animal

All animal experimental protocols were approved by the Animal Care and Use Committee (ACUC) at Fudan University (Shanghai, China). Male C57BL/6J mice, aged 6 weeks, were obtained from the Charles River Labs. Homozygotes of the whole-body Tet2 knockout (Tet2 KO) strain were originally

purchased from the Jackson Laboratory (Jackson Laboratories, Bar Harbor, ME, USA, stock no. #:023359). These mice were housed under specific pathogen-free conditions at 25°C with a 12 hr light/dark cycle and were fed either a standard chow or an HFD (Research Diets, 45% calories from fat). To induce insulin resistance, WT C57BL/6J mice aged 6 weeks were placed on an HFD for 11 days or 12 weeks. For the liver-specific *Tet2*-knockdown mice, we transduced AAV8 shRNA against *Tet2* to knock down *Tet2* in the liver (designed and synthesized by Hanbio, Shanghai, China) of mice. Scramble shRNA (AAV8) was used as a negative control. Target sequences for *Tet2* shRNA were 5'-GGAAUAUC CCACAUGAAAGGCAGCC-3'. In brief, male C57BL/6J mice aged 6 weeks were switched from standard chow to HFD diet. Mice were then randomly divided into four groups: AAV8, AAV8+metformin, AAV8-*shTet2*, AAV8-*shTet2*+metformin. AAV8 and AAV8-*shTet2* were diluted with saline to $1\times10^{12}$ vector genomes/mL, and 100 µL was injected through the tail vein for each mouse. At the end of the treatment, mice were euthanized, and liver tissues were collected for the subsequent immunofluorescence and protein extraction.

## Statistical analysis

Sample numbers are indicated in the figures and figure legends. The Student's t-test was used to determine the significant difference between two groups, and one-way analysis of variance (ANOVA) with Tukey's post hoc test for multiple comparisons when more than two groups were analyzed. Data analysis was performed using SPSS version 16.0 (SPSS Inc, Chicago, IL, USA) or GraphPad Prism (version 8.0.2; GraphPad Software, Inc, San Diego, CA, USA). A p-value of less than 0.05 was considered statistically significant. *, **, ***, and **** indicated statistically significant results compared to the control and represented $p<0.05$, $p<0.01$, $p<0.001$, and $p<0.0001$, respectively.

## Acknowledgements

This work was supported by the National Key R&D Program of China (2022YFA0807100, 2020YFA0803400/2020YFA0803402), the National Natural Science Foundation of China (82372754, 82472850, 82172936, 81972620, 82121004, and 82073128), the Shanghai Rising-Star Program (24QA2709900), the Program for Professor of Special Appointment (Eastern Scholar) at Shanghai Institutions of Higher Learning and the Fundamental Research Funds for the Central Universities.

# Additional information

### Funding

| Funder | Grant reference number | Author |
| --- | --- | --- |
| National Natural Science Foundation of China | 82372754 | Yanping Xu |
| National Key Research and Development Program of China | 2022YFA0807100 | Yanping Xu |
| National Natural Science Foundation of China | 82472850 | Lei Lv |
| National Natural Science Foundation of China | 82172936 | Lei Lv |
| National Natural Science Foundation of China | 81972620 | Lei Lv |
| National Natural Science Foundation of China | 82121004 | Lei Lv |

| Funder | Grant reference number | Author |
|---|---|---|
| National Natural Science Foundation of China | 82073128 | Yanping Xu |
| Shanghai Rising-Star Program | 24QA2709900 | Yanping Xu |
| National Key Research and Development Program of China | 2020YFA0803400/2020YFA0803402 | Yanping Xu |

The funders had no role in study design, data collection and interpretation, or the decision to submit the work for publication.

#### Author contributions
Hongchen Li, Conceptualization, Writing – original draft; Xinchao Zhang, Xiaoben Liang, Conceptualization, Data curation, Supervision, Funding acquisition, Writing – review and editing; Shuyan Li, Conceptualization, Formal analysis, Writing – original draft; Ziyi Cui, Writing – original draft; Xinyu Zhao, Methodology, Writing – original draft; Kai Wang, Data curation, Writing – review and editing; Bingbing Zha, Data curation, Formal analysis; Haijie Ma, Ming Xu, Conceptualization, Data curation, Formal analysis; Lei Lv, Conceptualization, Supervision, Funding acquisition, Writing – review and editing; Yanping Xu, Conceptualization, Data curation, Supervision, Funding acquisition

#### Author ORCIDs
Hongchen Li ⬤ http://orcid.org/0000-0003-2639-8726
Ming Xu ⬤ https://orcid.org/0000-0002-4477-939X
Lei Lv ⬤ https://orcid.org/0000-0003-4820-3125
Yanping Xu ⬤ https://orcid.org/0000-0003-2750-5466

#### Ethics
All animal experiments were approved by the Animal Care and Use Committee at Fudan University (Shanghai, China).

Reviewer #2 (Public review): https://doi.org/10.7554/eLife.103663.3.sa1
Author response https://doi.org/10.7554/eLife.103663.3.sa2

## Additional files

#### Supplementary files
MDAR checklist

#### Data availability
All data generated or analyzed during this study are included in the manuscript; source data files have been provided for Figures 1, 3, 4, 5 and 6.

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
