## [Editor Report · eLife Assessment]

Zhang et al. present **important** findings that reveal a new role for TET2 in controlling glucose production in the liver, showing that both fasting and a high-fat diet increase TET2 levels, while its absence reduces glucose production. TET2 works with HNF4α to activate the FBP1 gene upon glucagon stimulation, while metformin disrupts TET2-HNF4α interaction, lowering FBP1 levels and improving glucose homeostasis. The results are **convincing** and expand our understanding of gluconeogenesis regulation.

---

## [Referee Report · Reviewer #2 (Public review)]

The manuscript "HNF4α-1 TET2-FBP1 axis contributes to gluconeogenesis and type 2 diabetes" from Zhang et al. presents significant and convincing findings that enhance our understanding of TET2's role in liver glucose metabolism. It highlights the epigenetic regulation of FBP1, a gluconeogenic gene, by TET2, linking this pathway to HNF4alpha which recruits TET2. The in vitro and in vivo experiments are now well-described and provide convincing evidence of TET2's impact on gluconeogenesis, particularly in fasting and HFD mice.

Comments on revisions:

The authors have thoroughly addressed all the concerns raised, and their responses adequately clarify the issues previously identified.

Minor changes:

(1) Could the authors provide some comments on why glucagon was not able to stimulate PEPCK and G6Pase mRNA levels in HepG2 cells (Fig. 3D)? Although it is not the focus of the research, it is well known that glucagon has this effect and could serve as a positive control for the quality of the preparation.

(2) Please include the sequences of the qPCR primers used for PEPCK and G6Pase in the Methods section (page 17).

---

## [Author Response]

The following is the authors’ response to the original reviews

**Public Reviews:**

**Reviewer #1 (Public review):**
Summary:Zhang et al. describe a delicate relationship between Tet2 and FBP1 in the regulation of hepatic gluconeogenesis.Strengths:The studies are very mechanistic, indicating that this interaction occurs via demethylation of HNF4a. Phosphorylation of HNF4a at ser 313 induced by metformin also controls the interaction between Tet2 and FBP1.

We are grateful for the reviewer's praise on the manuscript.

Weaknesses:The results are briefly described, and oftentimes, the necessary information is not provided to interpret the data. Similarly, the methods section is not well developed to inform the reader about how these experiments were performed. While the findings are interesting, the results section needs to be better developed to increase confidence in the interpretation of the results.

Thanks very much for pointing out the shortcomings of the manuscript. We apologize that we did not provide detailed description for some experimental methods and results. Following reviewer’s suggestion, we added the details in method section, including the generation of whole-body *Tet2 KO* mice and liver-specific Tet2 knockdown mice (AAV8-shTet2), the missing information of reagent, antibody, primer sequences and mutant generation, and the methods of chromatin immunoprecipitation (ChIP) and immunofluorescence. The interpretation of the results was also further developed according to reviewer’s comments.

**Reviewer #2 (Public review):**
Summary:This study reveals a novel role of TET2 in regulating gluconeogenesis. It shows that fasting and a high-fat diet increase TET2 expression in mice, and TET2 knockout reduces glucose production. The findings highlight that TET2 positively regulates FBP1, a key enzyme in gluconeogenesis, by interacting with HNF4α to demethylate the FBP1 promoter in response to glucagon. Additionally, metformin reduces FBP1 expression by preventing TET2-HNF4α interaction. This identifies an HNF4α-TET2-FBP1 axis as a potential target for T2D treatment.Strengths:The authors use several methods in vivo (PTT, GTT, and ITT in fasted and HFD mice; and KO mice) and in vitro (in HepG2 and primary hepatocytes) to support the existence of the HNF4alpha-TET-2-FBP-1 axis in the control of gluconeogenesis. These findings uncovered a previously unknown function of TET2 in gluconeogenesis.

We are grateful for the reviewer's praise on the manuscript.

Weaknesses:Although the authors provide evidence of an HNF4α-TET2-FBP1 axis in the control of gluconeogenesis, which contributes to the therapeutic effect of metformin on T2D, its role in the pathogenesis of T2D is less clear. The mechanisms by which TET2 is up-regulated by glucagon should be more explored.

Thanks very much for pointing out the shortcomings of the manuscript. We agree with the reviewer that the manuscript is focused on the function of HNF4α-TET2-FBP1 axis in the control of gluconeogenesis, but not on its role in the pathogenesis of T2D. Following reviewer’s suggestion, we changed the title of the manuscript to “HNF4α-TET2-FBP1 axis contributes to gluconeogenesis and type 2 diabetes”. For the mechanisms by which TET2 is up-regulated by glucagon, we examined TET2 mRNA levels at different time points after a single dose of glucagon treatment in HepG2 cells. Interestingly, the results showed that TET2 mRNA levels significantly increased by 6 folds at 30 min and the sustained effect of glucagon on Tet2 mRNA levels persisted for more than 48 hours (refer to Fig. 3E).

**Recommendations for the authors:**

**Reviewer #1 (Recommendations for the authors):**
The authors indicate that they have overexpressed TET2 in HepG2 cells and primary mouse hepatocytes. The degree of overexpression should be shown. Is this similar to an increase in TET2 with fasting or HFD treatment?

Thanks for reviewer’s helpful comment. Following reviewer’s suggestion, we examined the protein levels of overexpressed TET2 in HepG2 cells and primary mouse hepatocytes. The results revealed that the degree of TET2 overexpression (refer to Fig. 3J) is similar to the increase of TET2 under fasting or HFD treatment (Fig. 1C, D).

In Figures 2E-2G, the authors report results in Tet2-KO mice. Information on how these mice were generated is lacking. There is limited information about how Tet2-KO cells were generated, but again, I could not find anything about these mice in the methods section or figure legend. Is this whole-body or liver-specific Tet2-KO? How old were the mice at the time of PTT, GTT, or ITT?Were these mice on chow or HFD? Are there any differences in body weight between WT and Tet2-KO mice?

Thanks for reviewer’s helpful comment. Following reviewer’s suggestion, we provided the detailed information about the *Tet2-KO* mice, including the mouse generation in methods section. Moreover, the details of *Tet2-KO* mice used in each figure were clearly described in the figure legend. In this study, two mouse models were employed: whole-body *Tet2-KO* mice and liver-specific TET2 knockdown mice (AAV8-shTet2). The mice used for PTT, GTT and ITT were 8 weeks old and on HFD. To address reviewer’s concern, we compared the body weight of WT and *Tet2-KO* mice and results revealed that no significant differences in the body weight between WT and *Tet2-KO* mice at 8 and 10 weeks old when on a normal chow diet, as depicted in Figure 2I.

Figures 3A-C shows that 48 hours after glucagon treatment, Tet2 and FBP1 mRNA increased. It's surprising that a single dose of glucagon would have effects that last that long. The peak rise in glucose following glucagon treatment occurs in 30 minutes. How do authors explain such a long effect of glucagon on Tet2 mRNA and protein?

Thanks for reviewer’s constructive comment. To address reviewer’s concern, we examined the mRNA levels of TET2 and FBP1 at different time points following a single dose of glucagon treatment in HepG2 cells. Interestingly, the results showed that TET2 mRNA levels significantly increased by 6 folds at 30 min and the sustained effect of glucagon on Tet2 mRNA levels persisted for more than 48 hours (refer to Fig. 3E). The detailed mechanism underlying long effect of glucagon on Tet2 mRNA and protein needs further exploration.

It's interesting that in Figure 3F, Fbp1 and Tet2 mRNA expression correlated positively in both ad libitum and fasting conditions. I would expect that during fed conditions, gluconeogenesis would not be activated and thus would expect no correlation.

Thanks for reviewer’s constructive comment. According to the results in new Fig. 3H, the mRNA levels of Fbp1 and Tet2 indeed positively correlated in both ad libitum and fasting conditions, while the r value is higher and p value is lower in fasting condition compared to ad libitum. Notably, both the expression levels of Fbp1 and Tet2 increased under fasting treatment, which is consistent with Fig. 1C and Fig. 4K.

The authors state that "Our results demonstrated that HNF4α recruits TET2 to the FBP1 promoter and activates FBP1 expression through demethylation" What data points out that this is mediated through demethylation?

Thanks for reviewer’s constructive comment. Following reviewer’s suggestion, we conducted new ChIP experiments. These data demonstrated that HNF4α recruits TET2 to the FBP1 promoter and activates FBP1 expression through demethylation, as showed in Fig. 4F-H.

For Figures 5B, 4D, and 3L-N y-axes are labeled as fold enrichment. The authors should clearly indicate what was being measured on y-axes.

Thanks for reviewer’s helpful comment. Following reviewer’s suggestion, we clearly labeled all the y-axes in each figure.

The authors indicate that metformin increases phosphorylation of Hnf4a at ser 313 Figure 5C. How do we know that ser 313 is involved? Only one antibody is listed for Hnf4a (SAB, 32591).

Thanks very much for pointing out. We determined the phosphorylation levels of HNF4α at S313 using Anti-HNF4α (phospho S313) (ab78356), we apologize for not labeling it clearly. Now, we made it clear in Fig. 5C and the detailed information of the antibody was added to the method section of “Western Blot and Immunoprecipitation”.

How did the authors make phosphomimetic mutation (S313D) and phosphoresistant mutation (S313A) of HNF4α? This is not described.

Thanks very much for pointing out. Following reviewer’s suggestion, the detailed method for making phosphomimetic mutation (S313D) and phosphoresistant mutation (S313A) of HNF4α was added to the method section of “Gene Knockout Cells and Mutagenesis”.

**Reviewer #2 (Recommendations for the authors):**
Major points:(1) Other key gluconeogenesis genes (e.g. PEPCK and G6Pase) should have been investigated to demonstrate whether or not the regulation of TET-2 is specific on FBP-1.

Thanks for reviewer’s helpful comment. Following reviewer’s suggestion, we designed the qPCR to assay other key gluconeogenesis genes, including PEPCK and G6Pase, and the results showed that glucagon treatment had no effect on PEPCK and G6Pase expression (Fig. 3D), suggesting the regulation of TET2 is specific on FBP1.

(2) The methods are not well defined and more details should be given, for example, to explain how the Tet2 KO mice were generated. Since these animals are not KO liver-specific and TET2 is expressed in a variety of tissues and organs and is predominantly found in hematopoietic cells, including bone marrow and blood cells, the phenotype of these mice should be better characterized.

Thanks for reviewer’s helpful comment. The *Tet2* knockout (*Tet2 KO*) mice were originally purchased from the Jackson Laboratory (strain No. 023359) and we added the detailed information to method section of “Animal”. According to the previously reported phenotype of *Tet2* KO mice, it mainly includes bone marrow, spleen, islet and heart. Specifically, Tet2 KO mice led to an increase of total cell numbers in the bone marrow and spleen (PMID: 21873190), as well as an elevated white blood cell (WBC) count (PMID: 37541212). Additionally, Tet2 KO mice exhibited splenomegaly (PMID: 37541212, PMID: 21723200, PMID: 38773071, PMID: 21723200). And the morphology of the islets (PMID: 34417463), anatomical chamber volumes or ventricular functions (PMID: 38357791) were indistinguishable between the *Tet2* KO and wild type (WT) mice.

(3) An experiment showing the co-localization of TET2 and HNF4α in the mouse liver in fasted mice and/or in HFD-mice would strengthen the data shown in Figure 3.

Thanks for reviewer’s helpful comment. Following reviewer’s suggestion, the experiments showing the co-localization of TET2 and HNF4α in the mouse liver in fasted mice and FD mice were conducted, as shown in new Fig. 4B and C.

Minor points:(1) Given that the manuscript does not focus on the role of TET2 in the pathogenesis of T2D, its title should be changed.

hanks for reviewer’s helpful comment. Following reviewer’s suggestion, we changed the title of the manuscript to “HNF4α-TET2-FBP1 axis contributes to gluconeogenesis and type 2 diabetes”.

(2) Please indicate the molecular weight of bands in all figures.

Thanks for reviewer’s helpful comment. Following reviewer’s suggestion, the molecular weight of bands was indicated in all figures.

(3) Why do the control values of the y-axis in Figure 1 A and B are so different? Please maintain the same scale in both figures.

Thanks for reviewer’s helpful comment. Following reviewer’s suggestion, we recalculated and normalized the control value in Fig. 1A to maintain the same scale in both figures.

(4) In Figure 2F, do the plasma insulin levels have altered in response to GTT in Tet2-KO mice? If so, please show the data and discuss.

Thanks for reviewer’s helpful comment. Following reviewer’s suggestion, we examined the plasma insulin levels in the process of GTT assay, and the result revealed that Tet2-KO mice showed lower insulin levels after glucose administration, which reflects higher insulin sensitivity, as shown in new Fig. 2H.

(5) The increase of TET2 hepatic protein levels in response to fasting occur in other tissues and hematopoietic cells?

Thanks for reviewer’s helpful comment. Following reviewer’s suggestion, we examined Tet2 protein levels under fasting condition in other tissues and hematopoietic cells, and found that fasting also increased Tet2 protein levels in kidney, brain, and hematopoietic cells, but not in heart.

**Author response image 1. sa2fig1:** 

(6) Please indicate the glucagon concentration and metformin dose in all figures in which they are mentioned.

Thanks for reviewer’s helpful comment. Following reviewer’s suggestion, the glucagon concentration (20 nM) and metformin concentration (10 mM for HepG2 cell treatment and 300 mg/kg per day for mice treatment) were added in the figure legends, respectively.